# Inelastic Processes in Strontium-Hydrogen Collisions and Their Impact on Non-LTE Calculations

Svetlana A. Yakovleva [1,*], Andrey K. Belyaev [1,*] and Lyudmila I. Mashonkina [2,*]

[1] Department of Theoretical Physics and Astronomy, Herzen University, 191186 St. Petersburg, Russia
[2] Institute of Astronomy of the Russian Academy of Sciences, Pyatnitskaya St. 48, 119017 Moscow, Russia
[*] Correspondence: sayakovleva@herzen.spb.ru (S.A.Y.); akbelyaev@herzen.spb.ru (A.K.B.); lima@inasan.ru (L.I.M.)

**Abstract:** Inelastic processes rate coefficients for low-energy Sr + H, $Sr^+ + H^-$, $Sr^+ + H$, and $Sr^{2+} + H^-$ collisions are calculated using the multichannel quantum model approach. A total of 31 scattering channels of $SrH^+$ and 17 scattering channels of SrH are considered. The partial cross sections and the partial rate coefficients are hence calculated for 1202 partial processes in total. Using new quantum data for Sr II + H I collisions, we updated the model atom of Sr II and performed the non-local thermodynamic equilibrium (non-LTE) calculations. We provide the non-LTE abundance corrections for the Sr II resonance lines in two grids of model atmospheres, which are applicable to very metal-poor ([Fe/H] $\leq -2$) dwarfs and giants.

**Keywords:** atomic collisional data; atomic inelastic processes; rate coefficients; stars: atmospheres

## 1. Introduction

The stars with a mass of less than the solar one have a long lifetime comparable with the Galaxy age, and they play an important role in providing observational material for the Galactic chemical evolution studies. One of the directions of such studies is a restoration of the synthesis history for heavy nuclei produced by the neutron-capture nuclear reactions.

In these reactions, neutrons are captured by nuclei that can then undergo β decay if they are unstable, transforming neutrons into protons and producing elements of the higher atomic number. Depending on the neutron flux available, which determines the lifetime for neutron capture, $\tau_n$, the neutron-capture mechanisms are separated into the slow (s-) processes and the rapid (r-) processes [1]. If $\tau_n$ is greater than the radioactive decay lifetime, $\tau_\beta$, for unstable nuclei, the process is referred to as a slow (s-) process, and the process is defined as a rapid (r-) process if $\tau_n \ll \tau_\beta$. Different isotopes have different yields from the neutron-capture nuclear reactions, either s- or r-process, with relatively sharp peaks at the magic numbers $N_n = 50$ and 82 on the abundance curve, $N_n$ being the number of neutrons. Strontium (Sr) is the best observed of the elements beyond the Fe group. The most abundant isotopes of strontium, $^{88}Sr$, $^{87}Sr$, and $^{86}Sr$, have $N_n = 50$, 49, and 48; thus, strontium is related to the first peak. The best representative of the second peak elements is barium (Ba). In the solar system matter, abundances of both Sr and Ba are mostly contributed from the main s-process occurring during the thermally pulsing asymptotic giant branch (AGB) phase of low mass ($1.3 \leq M/M_\odot < 4$) stars, with fractions of s-nuclei of about 70 and 85%, respectively [2]. In the early Universe, before the onset of the main s-process, the r-process is expected to be the only source of the neutron-capture elements. With observations of barium and the heavier elements in very metal-poor (VMP, [Fe/H] $\leq -2$), old stars provide evidence for their r-process origin [3]. The origin of the first peak elements and strontium among them is less clear.

If Ba and Sr were produced by the same nucleosynthesis source and before the onset of the main s-process, this would result in a fairly flat [Sr/Ba] ratio versus [Ba/H] for VMP stars. This is not supported by the observations. For the Galactic halo stars,

McWilliam et al. [4], Honda et al. [5], and François et al. [6] find increasing [Sr/Ba] ratio toward lower [Ba/H]. Similar tight anti-correlation of [Sr/Ba] with [Ba/H] is found by Mashonkina et al. [7] for the VMP stars in not only our Galaxy but also its satellites—classical dwarf spheroidal galaxies (dSphs). The latter paper reports also on the second group of stars, which reveal similar [Sr/Ba] $\sim -0.5$, in line with the empirical r-process ratio, $[\text{Sr/Ba}]_\text{r} = -0.44 \pm 0.08$ [8]. Thus, observations of VMP stars suggest that, in the early Universe, there existed the second nucleosynthesis channel for Sr, besides the classical r-process.

The extra source(s) of Sr is (are) not identified yet despite various ideas and nucleosynthesis models which have been proposed in the literature; see the most recent paper by Rizzuti et al. [9] and references therein. Further progress can be provided by improving observational constraints to the theoretical models. Therefore, one needs to improve an accuracy of the derived stellar abundances of Sr and Ba, as well as to increase their statistics. In order to determine accurate elemental abundances, the theoretical spectra should be calculated based on the non-local thermodynamic equilibrium (non-LTE = NLTE) line formation. For barium, the non-LTE modelings using the most up-to-date methods and atomic data available so far, including the quantum rate coefficients calculated by Belyaev and Yakovleva [10] for the inelastic processes in Ba I Ba II + H I and Ba II Ba III + H$^-$ collisions, were already treated [11,12]. The present paper is devoted to strontium.

In VMP stars, strontium is observed in lines of Sr II. The non-LTE methods for Sr II were developed by Belyakova and Mashonkina [13], Andrievsky et al. [14], and Bergemann et al. [15]. They all treat the H I impact excitation and de-excitation processes using the Steenbock and Holweger [16] formulas, which are based on the work of Drawin [17]. Such an approach has been criticized for not providing a realistic description of the physics involved and overestimating the collision rates [18]. This paper presents new quantum calculations of the Sr I + H I , Sr II + H I , Sr II + H$^-$, and Sr III + H$^-$ collisions. The obtained rate coefficients for the excitation and charge exchange processes by hydrogen impact were implemented in the model atom of Sr II. The model atom from Belyakova and Mashonkina [13] was taken as a basic model.

In the following, Section 2 gives the description of the atomic data calculations. Two quasimolecules are considered: ionic SrH$^+$ and neutral SrH ones. The former collisional system is treated in the J-J representation, that is, it includes the fine structure being taken into account, while the latter is in the L-S representation, that is, without accounting for the fine structure. The brief description of the way to treat a nonadiabatic nuclear dynamics is presented. The results of the quantum calculations for the inelastic processes of neutralization, ion-pair formation, excitations, and de-excitation are discussed as well. Section 3 describes the updated model atom of Sr II and presents the non-LTE abundance corrections for the Sr II 4077, 4215 Å resonance lines in the two grids of model atmospheres, which are applicable to VMP dwarfs and giants. Section 4 concludes the article.

## 2. Atomic Data Calculations

### 2.1. Processes in Sr$^+$ + H and Sr$^{2+}$ + H$^-$ Collisions

The electronic structure of a SrH$^+$ quasimolecule was studied with ab initio methods by Aymar and Dulieu [19] and more thoroughly by Mejrissi et al. [20], although none of these studies were performed in a relativistic approximation and have any information about the fine structure levels of a SrH$^+$ quasimolecule.

In order to study inelastic processes in collisions of Sr$^+$ with hydrogen atoms taking fine structure levels of Sr$^+$ into account, nonadiabatic nuclear dynamics should be investigated in the J-J representation. The ionic molecular state Sr$^{2+}(4p^6\ {}^1S_0) + H^-(1s^2\ {}^1S_0)$ has the same structure as in the case of alkali–hydrogen collisions: both partners have closed electronic shells and the ionic molecular state has the $0^+$ symmetry. For such collisions, an approach for taking fine structure of alkali atoms into account in quantum model calculations was proposed in [21]. This model suggests to use the asymptotic approach for calculations of diabatic potentials for $A^{Z+}({}^1S_0) + H^-({}^1S_0)$ and $A^{(Z-1)+}({}^2L_j) + H({}^2S_{1/2})$

states. The off-diagonal diabatic Hamiltonian matrix elements for the fine structure levels are calculated dividing semiempirical matrix elements [22] proportional to the coefficients:

$$
C = \frac{1}{\sqrt{2}}\left( \begin{bmatrix} j & {}^{1}/_{2} & J \\ {}^{1}/_{2} & -{}^{1}/_{2} & 0 \end{bmatrix} \begin{bmatrix} L & {}^{1}/_{2} & j \\ 0 & {}^{1}/_{2} & {}^{1}/_{2} \end{bmatrix} \right.
$$
$$
\left. - \begin{bmatrix} j & {}^{1}/_{2} & J \\ -{}^{1}/_{2} & {}^{1}/_{2} & 0 \end{bmatrix} \begin{bmatrix} L & {}^{1}/_{2} & j \\ 0 & -{}^{1}/_{2} & -{}^{1}/_{2} \end{bmatrix} \right),
\tag{1}
$$

where the square brackets denote the Clebsch–Gordan coefficients.

This approach for electronic structure modeling allows one to take into account nonadiabatic transitions due to the interaction of the ionic and covalent configurations of a quasimolecule [23]. For this reason, only covalent states of the $0^+$ molecular symmetry are taken into account. The calculations of the SrH$^+$ electronic structure are performed for 31 scattering channels listed in Table A1: 30 covalent molecular states and 1 ionic. The nonadiabatic nuclear dynamics is investigated using the multichannel Landau–Zener approach described in detail in [24,25]; the cross sections and rate coefficients are calculated for all inelastic processes due to transitions between these states.

In a multichannel case, a state-to-state probability for a transition from an initial channel $i$ to a final channel $f$ for the neutralization and de-excitation processes $(i > f)$ is calculated by the following equations

$$
P_{if}^{neutr} = 2p_f(1 - p_f)\left( \prod_{k=f+1}^{i-1} p_k \right)\left\{ 1 + \sum_{m=1}^{2(f-1)} \prod_{k=1}^{m}\left( -p_{f-\left[\frac{k+1}{2}\right]} \right) \right\},
$$
$$
P_{if}^{deex} = 2p_f(1 - p_f)(1 - p_i)\left( \prod_{k=f+1}^{i-1} p_k \right)\left\{ 1 + \sum_{m=1}^{2(f-1)} \prod_{k=1}^{m}\left( -p_{f-\left[\frac{k+1}{2}\right]} \right) \right\},
\tag{2}
$$

where the nonadiabatic transition probability $p_j$ after a single passing of a nonadiabatic region is calculated by means of the two-state Landau–Zener model [26–28] by the following formula

$$
p_j = \exp\left( -\frac{\xi_j}{v} \right),
\tag{3}
$$

the Landau–Zener parameter $\xi_j$ being computed by means of the adiabatic-potential based formula derived by Belyaev and Lebedev [29]

$$
\xi_j = \frac{\pi}{2\hbar}\sqrt{\left.\frac{Z_j^3}{Z_j''}\right|_{R=R_C}}.
\tag{4}
$$

In this equation, $Z_j$ is the absolute value of the adiabatic potential splitting between neighboring states: $Z_j(R) = |U_j(R) - U_{j\pm1}(R)|$, $R_c$ being a center of a nonadiabatic region, $R''$ denotes the second derivative with respect to the internuclear distance $R$. The latter formula allows one to calculate the nonadiabatic transition probability only from the information about the adiabatic potentials without making transformation into a diabatic representation which is not uniquely defined and time-consuming. The partial cross sections and the partial rate coefficients are calculated from the state-to-state transition probabilities as usual, see, e.g., Reference [23].

Let us discuss the results of the rate coefficients calculations. The graphical representation of the calculated rate coefficients for the temperature T = 6000 K is shown in Figure 1, where rate coefficients are presented by color from blue to red according to the legend. One can see from Figure 1 that the rate coefficients with the highest values ($\geq 10^{-8}$ cm$^3$/s) correspond to the neutralization processes to the following final states of a strontium ion: Sr$^+\left(4p^66s\,^2S_{1/2}\right)$, Sr$^+\left(4p^65d\,^2D_{3/2,5/2}\right)$, Sr$^+\left(4p^66p\,^2P_{1/2,3/2}\right)$,

$Sr^+\left(4p^64f\ ^2F_{7/2,5/2}\right)$ and $Sr^+\left(4p^67s\ ^2S_{1/2}\right)$ (the scattering channels k = 6–13 in Table A1). At T = 6000 K, the rate coefficients have the values from $1.36 \times 10^{-8}$ to $3.48 \times 10^{-8}$ cm³/s. Several processes have rate coefficients with moderate values (from $10^{-10}$ to $10^{-8}$ cm³/s): mainly these are other neutralization processes and excitation/de-excitation processes associated with transitions between states k = 6–13 from Table A1.

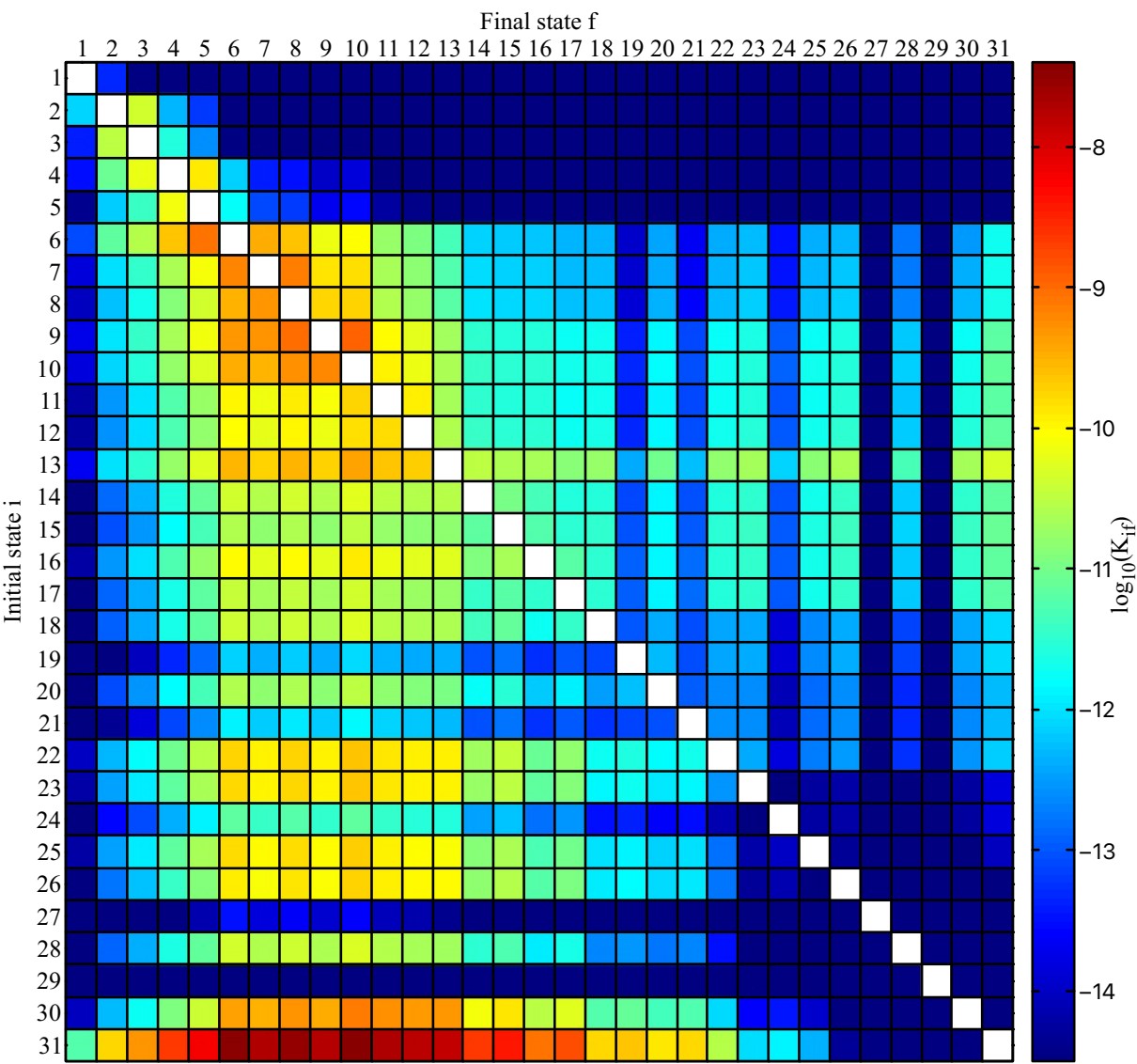

**Figure 1.** The graphical representation for the inelastic processes rate coefficients in $Sr^+\left(4p^6nl\ ^2L_j\right)$ + $H(1s\ ^2S_{1/2})$ and $Sr^{2+}\left(4p^6\ ^1S_0\right)$ + $H^-\left(1s^2\ ^1S_0\right)$ collisions at the temperature T = 6000 K. The labels for the initial and final states are given in Table A1.

Figure 2 shows the dependence of the neutralization rate coefficients on the excitation energy of the final state calculated both with and without account for the fine structure levels of $Sr^+$. The LS and JJ calculations give almost the same rate coefficients for neutralization processes to $Sr^+\left(4p^6ns\ ^2S_{1/2}\right)$ + H states that have only one fine structure level, while the results for the neutralization processes to other states show non-trivial behavior. Rate coefficients for the processes to fine structure levels cannot be obtained by dividing the results of LS calculations proportionally to the statistical populations of the final states or even proportionally to the coefficients given by Equation (1). This is due to the reason that each nonadiabatic region is changed differently with the transition of the L-S representation

to the J-J representation. This fact has different effects on the transition probabilities and eventually on the rate coefficients.

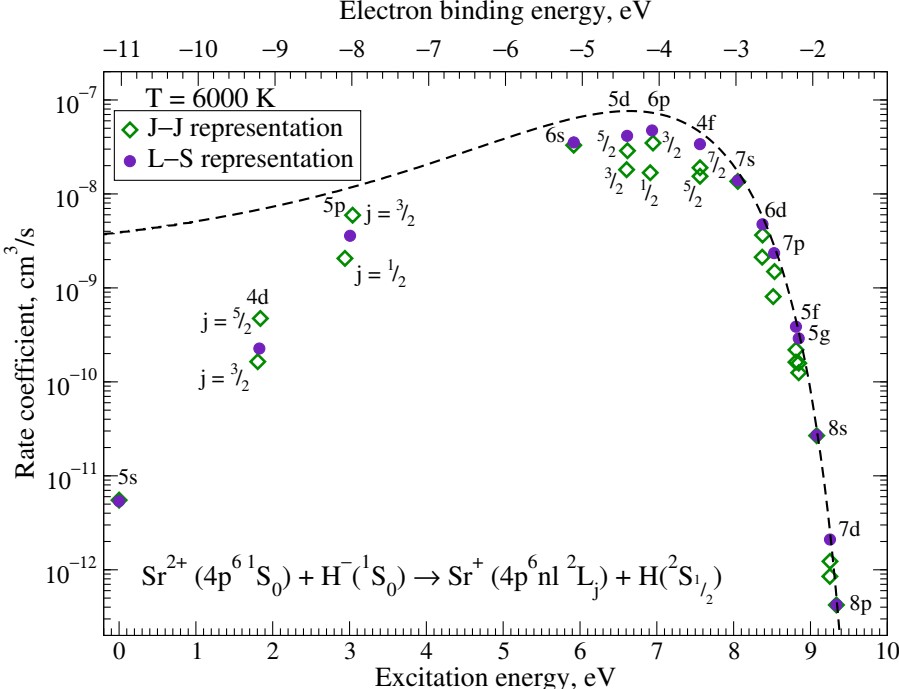

**Figure 2.** The rate coefficients (in $cm^3/s$) at T = 6000 K for the neutralization processes in $Sr^{2+}\left(4p^6\ ^1S_0\right) + H^-\left(1s^2\ ^1S_0\right)$ collisions. Empty diamonds correspond to the calculations with the account for the fine structure; filled circles, to the calculations without fine structure; and dashed line shows the general dependence of the rate coefficient according to the simplified quantum model [30].

## 2.2. Processes in Sr + H and Sr$^+$ + H$^-$ Collisions

Calculations of rate coefficients for the inelastic processes in collisions of Sr atoms with hydrogen atoms are performed in L-S representation, that is, without account for the fine structure levels of Sr. Same as for the case of SrH$^+$, we applied the asymptotic approach for SrH electronic structure modeling [23] and multichannel analytical Landau–Zener formulae for nonadiabatic nuclear dynamics [24,25]. Ionic molecular state $Sr^+\left(5s\ ^2S\right) + H^-\left(1s^2\ ^1S\right)$ has $^2\Sigma^+$ molecular symmetry; for this reason, the investigation is performed within this symmetry. A total of 17 scattering channels that form $^2\Sigma^+$ molecular states are considered and collected in Table A2 together with their asymptotic energies taken from NIST database [31]. Cross sections and rate coefficients are calculated for all inelastic processes due to the transitions between the states in Table A2.

Figure 3 shows the graphical representation of the calculated rate coefficients in SrH collisions. Only three inelastic processes have rate coefficients higher than $10^{-8}\ cm^3/s$: two mutual neutralization processes $Sr^+\left(5s\ ^2S\right) + H^-\left(1s^2\ ^1S\right) \rightarrow Sr\left(5s6s\ ^{1,3}S\right) + H(1s\ ^2S)$ and one de-excitation process $Sr\left(5s6s\ ^1S\right) + H(1s\ ^2S) \rightarrow Sr\left(5s6s\ ^3S\right) + H(1s\ ^2S)$. At the temperature T = 6000 K, the corresponding values of the rate coefficients are $(4.92, 5.39, 1.32) \times 10^{-8}\ cm^3/s$.

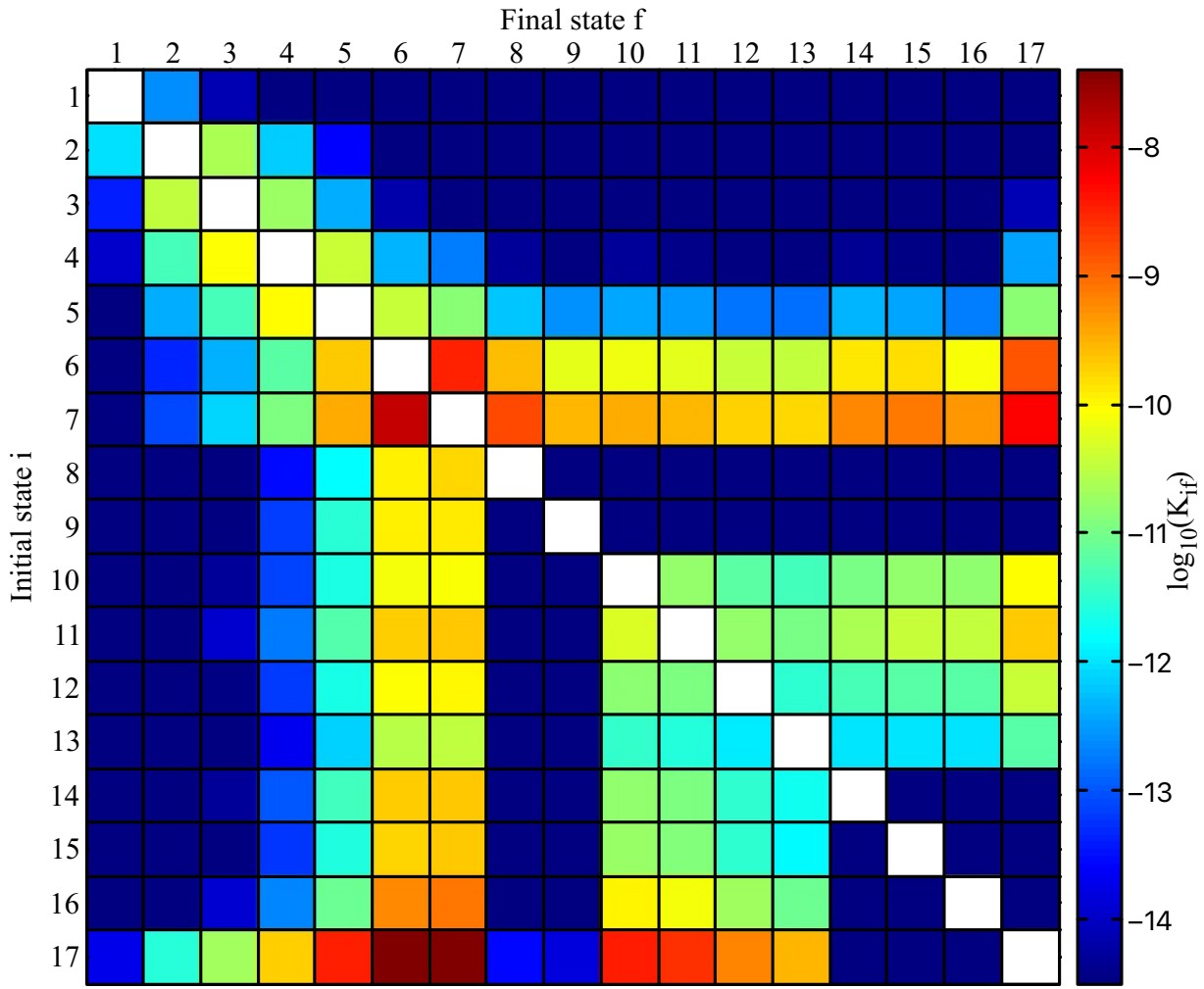

**Figure 3.** The graphical representation for the inelastic processes rate coefficients in $Sr(^{1,3}L) + H(1s\,^2S)$ and $Sr^+(5s\,^2S) + H^-(1s^2\,^1S)$ collisions. The labels for the initial and final states are given in Table A2.

The mutual neutralization rate coefficients at T = 6000 K are presented in Figure 4 as a function of the excitation energy of the final state. One can see from Figure 4 that rate coefficients of the processes due to the one-electron transitions are in good agreement with the general dependence of the rate coefficients given by the simplified quantum model [32] which is plotted by the dashed line. The deviations from this behavior at low excitation energies occur due to the presence of the long-range nonadiabatic regions. In this case, the multichannel approach gives smaller values for the transition probabilities to the lower-lying states than the two-state approximation used in the simplified quantum model. This general dependence allows one to evaluate the optimal window for the energies of the scattering channels that are likely to be initial or final channels of the processes with high values of the rate coefficients. As for the processes due to the two-electron transitions, they cannot be considered with the simplified quantum model and are treated differently in electronic structure calculations. Nonadiabatic regions that correspond to two-electron transitions are usually very narrow, even if the excitation energy of the covalent state lies in the optimal window, and that leads to small values of the transition probabilities in such regions.

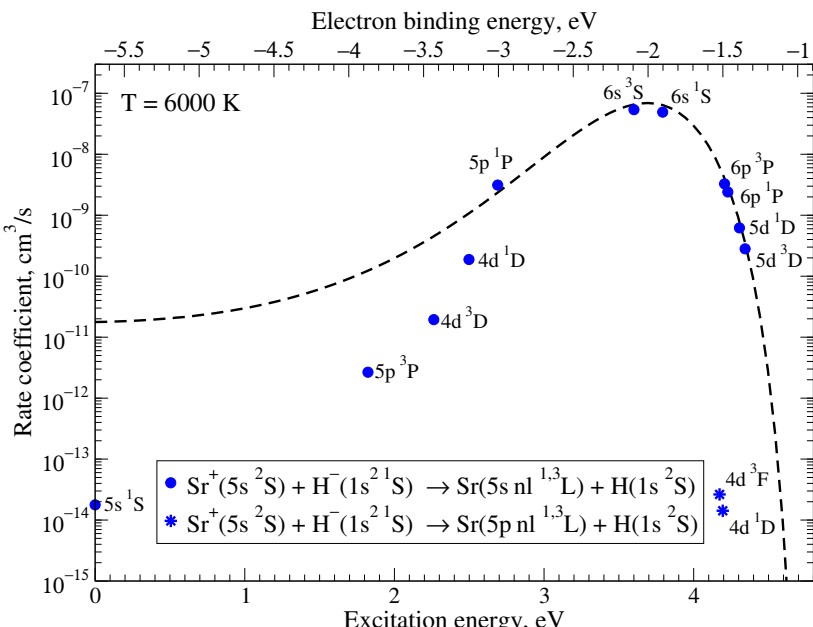

**Figure 4.** The rate coefficients (in cm$^3$/s) at T = 6000 K for the mutual neutralization processes in Sr$^+$ (5$s$ $^2$S) + H$^-$ (1$s^2$ $^1$S) collisions. Filled circles correspond to the processes due to one-electron transitions; stars, to two-electron transitions; dashed line shows the general dependence of the rate coefficient according to the simplified quantum model [32].

## 3. Non-LTE Calculations for Sr II

### 3.1. Model Atom of Sr II and Method of Calculations

We use a comprehensive model atom of Sr II that was constructed by Belyakova and Mashonkina [13] and updated later by including electron-impact excitation rate coefficients from ab initio calculations of Bautista et al. [33]. In this study, we implemented the H I impact excitation and de-excitation processes in Sr II, with the rate coefficients described in Section 2.

We briefly describe the atomic data we used. The model atom includes 40 levels, up to 12d, of Sr II and the ground state of Sr III. The transition probabilities were taken from Reader et al. [34], Lindgård and Nielson [35], and R. Kurucz's website[1]. Photoionization cross sections for $n$s, $n$p, and $n$d levels have been calculated by the quantum defect method using Peach [36] tables. For the remaining levels, we applied the hydrogenic approximation, where a principal quantum number $n$ was replaced with an effective principal quantum number $n_{\text{eff}}$. We note that, for effective temperatures of $T_{\text{eff}} > 4000$ K, Sr II is the majority species in the line formation layers, and the uncertainty in the adopted photoionization cross sections affects only weakly the statistical equilibrium (SE) of Sr II. Bautista et al. [33] provide accurate electron-impact excitation rate coefficients for all the transitions between 5s and 4f. The formula of van Regemorter [37] for allowed transitions and the effective collision strength Y = 1 for forbidden transitions were used in the other cases. Ionization by electronic collisions was calculated from the Seaton [38] formula, employing a hydrogenic photoionization cross section at the threshold. The H I impact excitation and de-excitation processes were taken into account for all the transitions between 5s and 9s, using accurate rate coefficients from quantum calculations (Section 2).

To solve the coupled radiative transfer and SE equations, we employed the DETAIL code [39], where the opacity package was modified, as described by Mashonkina et al. [40]. The calculations were performed with the MARCS plane-parallel (1D) model atmospheres with standard abundances [41] available on the MARCS website[2].

### 3.2. Non-LTE Effects for Lines of Sr II

In VMP stars, strontium can only be observed in the Sr II 4077 and 4215 Å resonance lines. We used $\log gf(4077\,\text{Å}) = 0.15$ and $\log gf(4215\,\text{Å}) = -0.17$, as recommended by the National Institute of Standards and Technology (NIST) database[3] [31], and $\log \Gamma_6/N_\text{H} = -7.71$ from Barklem and O'Mara [42]. Here, the van der Waals broadening constant $\Gamma_6$ corresponds to a temperature of 10,000 K and $N_\text{H}$ is the number density of neutral hydrogen.

Figure 5 displays the non-LTE abundance corrections, $\Delta_\text{NLTE} = \log \varepsilon_\text{NLTE} - \log \varepsilon_\text{LTE}$, for Sr II 4077 Å in the model atmospheres representative of VMP giants ($T_\text{eff}$ = 4500 K) and dwarfs ($T_\text{eff}$ = 5250 K). As found first by Belyakova and Mashonkina [13], the dominant non-LTE mechanisms for the Sr II resonance lines change, when moving from close-to-solar to the lower metallicity, resulting in a change of negative by positive $\Delta_\text{NLTE}$. The metallicity, where $\Delta_\text{NLTE}$ changes its sign, depends on $T_\text{eff}$, surface gravity log g, and the Sr abundance. This is seen in Figure 5 for the $T_\text{eff}$ = 5250 K models. The non-LTE abundance corrections are negative in the models, where Sr II 4077 Å is strong and its core forms in the uppermost atmospheric layers. At these depths, the upper level, $5p\,^2\text{P}^\circ_{3/2}$, of the resonance transition is underpopulated because of photon losses in the resonance lines, while the ground state keeps its thermodynamic equilibrium population, resulting in decreasing the line source function below the Planck function and an enhanced absorption in the line core. The non-LTE abundance correction for Sr II 4077 Å is positive, if the line forms in deep atmospheric layers, where $5p\,^2\text{P}^\circ_{3/2}$ is overpopulated by pumping transitions arising from the ground term.

In general, positive $\Delta_\text{NLTE}$ grows toward lower metallicity and lower surface gravity. For given $T_\text{eff}$/log g/[Fe/H], $\Delta_\text{NLTE}$ is larger for the lower Sr abundance ([Sr/Fe]). For stellar samples, which cover a broad metallicity range and include both dwarfs and giants, neglecting the non-LTE effects for lines of Sr II leads to distorted trends of [Sr/Fe] and [Sr/Ba] with metallicity and increased scatter of data for stars of close metallicity.

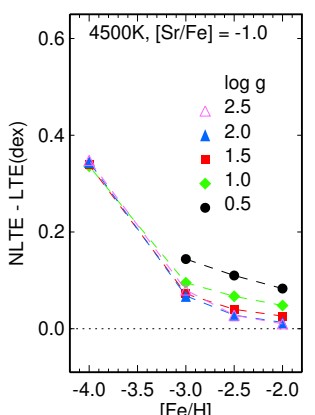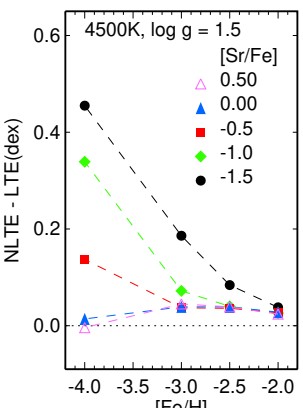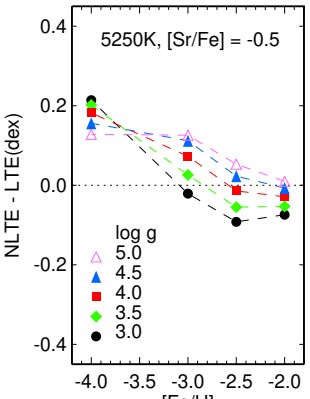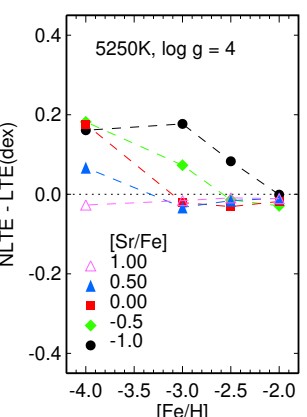

**Figure 5.** Non-LTE abundance corrections for Sr II 4077 Å in the model atmospheres of giants with common $T_\text{eff}$ = 4500 K and varied log g and [Sr/Fe] (two left panels) and dwarfs with common $T_\text{eff}$ = 5250 K and varied log g and [Sr/Fe] (two right panels) as a function of metallicity.

We computed non-LTE abundance corrections for the Sr II 4077 and 4215 Å lines in two grids of the MARCS model atmospheres, which are applicable to VMP dwarfs and giants. For each node ($T_\text{eff}$/log g/[Fe/H]) of the grids, the calculations were performed with five [Sr/Fe] values. Everywhere, a microturbulence velocity of 1.5 km s$^{-1}$ was adopted. The grids cover the following ranges of atmospheric parameters and Sr abundances.

Grid I: $T_\text{eff}$ = 5000 to 6500 K, with a step of 250 K; log g = 3.0 to 5.0, with a step of 0.5; [Fe/H] = $-2.0$, $-2.5$, $-3.0$, and $-4.0$; [Sr/Fe] = $-1.0$ to $+1.0$, with a step of 0.5.

Grid II: $T_\text{eff}$ = 4000 to 5000 K, with a step of 250 K; log g = 0.5 to 2.5, with a step of 0.5; [Fe/H] = $-2.0$, $-2.5$, $-3.0$, and $-4.0$; [Sr/Fe] = $-1.5$ to $+0.5$, with a step of 0.5.

The entire data set, together with the IDL routine for an interpolation in the grids for given $T_{\text{eff}}$, log g, [Fe/H], and observed line equivalent width, are available in machine-readable form[4].

## 4. Conclusions

Rate coefficients for excitation, de-excitation, ion-pair production, and neutralization processes in Sr + H, $Sr^+ + H^-$, $Sr^+ + H$, and $Sr^{2+} + H^-$ collisions are calculated using the quantum multichannel model approach [23–25]. Calculations are performed in the J-J representation for $SrH^+$, which allows us to consider 31 scattering channels taking fine structure of $Sr^+$ into account, and in the L-S representation for SrH considering 17 scattering channels. The inelastic cross sections and inelastic rate coefficients are calculated for 930 partial processes in Sr + H and $Sr^+ + H^-$ collisions, as well as for 272 partial processes in $Sr^+ + H$ and $Sr^{2+} + H^-$ collisions, for 1202 partial processes in total.

The Sr II model atom treated by Belyakova and Mashonkina [13] was updated by using new quantum mechanical rate coefficients for Sr II + H I and Sr II + H⁻ collisions. We performed the non-LTE calculations for Sr II with two grids of model atmospheres, which are applicable to VMP dwarfs and giants and provide the non-LTE abundance corrections for the Sr II 4077 and 4215 Å resonance lines. They are publicly available. The present non-LTE abundance corrections are based on the new atomic data for the inelastic processes in strontium–hydrogen collisions. As the result, the present data are more accurate than the old ones, and hence the non-LTE modeling is more reliable than the previous ones.

**Author Contributions:** Data calculations, S.A.Y.; data analysis, S.A.Y. and A.K.B.; data curation, A.K.B. and S.A.Y.; astrophysical applications, L.I.M.; writing, S.A.Y., A.K.B. and L.I.M. All authors have read and agreed to the published version of the manuscript.

**Funding:** S.A.Y. and A.K.B. gratefully acknowledge support from the Ministry for Education (Russian Federation), grant number FSZN-2020-0026.

**Data Availability Statement:** The calculated rate coefficients are available from the authors.

**Acknowledgments:** S.A.Y. and A.K.B. gratefully acknowledge support from the Ministry for Education and Science (Russian Federation).

**Conflicts of Interest:** The authors declare no conflict of interest.

## Abbreviations

The following abbreviations are used in this manuscript:

| | |
|---|---|
| LTE | Local thermodynamic equilibrium |
| NLTE | Non-local thermodynamic equilibrium |
| SN | Supernova |
| dSph galaxy | dwarf spheroidal galaxy |

## Appendix A

The scattering channels and the corresponding molecular states treated in the present work are collected in Tables A1 and A2 for the $SrH^+$ and SrH collisional systems.

**Table A1.** SrH$^+$ (k 0$^+$) molecular states (in the J-J representation), the corresponding scattering channels, their asymptotic energies with respect to the ground-state level (taken from NIST [31]).

| k | Scattering Channels | Asymptotic Energies (eV) |
|---|---|---|
| 1 | Sr$^+\left(4p^65s\,^2S_{1/2}\right)$ + H($1s\,^2S_{1/2}$) | 0.0 |
| 2 | Sr$^+\left(4p^64d\,^2D_{3/2}\right)$ + H($1s\,^2S_{1/2}$) | 1.8047016 |
| 3 | Sr$^+\left(4p^64d\,^2D_{5/2}\right)$ + H($1s\,^2S_{1/2}$) | 1.8394593 |
| 4 | Sr$^+\left(4p^65p\,^2P^\circ_{1/2}\right)$ + H($1s\,^2S_{1/2}$) | 2.9403088 |
| 5 | Sr$^+\left(4p^65p\,^2P^\circ_{3/2}\right)$ + H($1s\,^2S_{1/2}$) | 3.0396772 |
| 6 | Sr$^+\left(4p^66s\,^2S_{1/2}\right)$ + H($1s\,^2S_{1/2}$) | 5.9185754 |
| 7 | Sr$^+\left(4p^65d\,^2D_{3/2}\right)$ + H($1s\,^2S_{1/2}$) | 6.6066604 |
| 8 | Sr$^+\left(4p^65d\,^2D_{5/2}\right)$ + H($1s\,^2S_{1/2}$) | 6.6174048 |
| 9 | Sr$^+\left(4p^66p\,^2P^\circ_{1/2}\right)$ + H($1s\,^2S_{1/2}$) | 6.91456 |
| 10 | Sr$^+\left(4p^66p\,^2P^\circ_{3/2}\right)$ + H($1s\,^2S_{1/2}$) | 6.95029 |
| 11 | Sr$^+\left(4p^64f\,^2F^\circ_{7/2}\right)$ + H($1s\,^2S_{1/2}$) | 7.561801 |
| 12 | Sr$^+\left(4p^64f\,^2F^\circ_{5/2}\right)$ + H($1s\,^2S_{1/2}$) | 7.561962 |
| 13 | Sr$^+\left(4p^67s\,^2S_{1/2}\right)$ + H($1s\,^2S_{1/2}$) | 8.0545218 |
| 14 | Sr$^+\left(4p^66d\,^2D_{3/2}\right)$ + H($1s\,^2S_{1/2}$) | 8.3717688 |
| 15 | Sr$^+\left(4p^66d\,^2D_{5/2}\right)$ + H($1s\,^2S_{1/2}$) | 8.3767629 |
| 16 | Sr$^+\left(4p^67p\,^2P^\circ_{1/2}\right)$ + H($1s\,^2S_{1/2}$) | 8.515153 |
| 17 | Sr$^+\left(4p^67p\,^2P^\circ_{3/2}\right)$ + H($1s\,^2S_{1/2}$) | 8.532235 |
| 18 | Sr$^+\left(4p^65f\,^2F^\circ_{5/2}\right)$ + H($1s\,^2S_{1/2}$) | 8.811036 |
| 19 | Sr$^+\left(4p^65f\,^2F^\circ_{7/2}\right)$ + H($1s\,^2S_{1/2}$) | 8.811036 |
| 20 | Sr$^+\left(4p^65g\,^2G_{7/2}\right)$ + H($1s\,^2S_{1/2}$) | 8.847240 |
| 21 | Sr$^+\left(4p^65g\,^2G_{9/2}\right)$ + H($1s\,^2S_{1/2}$) | 8.847240 |
| 22 | Sr$^+\left(4p^68s\,^2S_{1/2}\right)$ + H($1s\,^2S_{1/2}$) | 9.080243 |
| 23 | Sr$^+\left(4p^67d\,^2D_{3/2}\right)$ + H($1s\,^2S_{1/2}$) | 9.251862 |
| 24 | Sr$^+\left(4p^67d\,^2D_{5/2}\right)$ + H($1s\,^2S_{1/2}$) | 9.254565 |
| 25 | Sr$^+\left(4p^68p\,^2P^\circ_{3/2}\right)$ + H($1s\,^2S_{1/2}$) | 9.337473 |

**Table A1.** *Cont.*

| k | Scattering Channels | Asymptotic Energies (eV) |
|---|---|---|
| 26 | $\mathrm{Sr}^+\left(4p^66f\ ^2\mathrm{F}^\circ_{7/2}\right) + \mathrm{H}(1s\ ^2\mathrm{S}_{1/2})$ | 9.491412 |
| 27 | $\mathrm{Sr}^+\left(4p^66f\ ^2\mathrm{F}^\circ_{5/2}\right) + \mathrm{H}(1s\ ^2\mathrm{S}_{1/2})$ | 9.491412 |
| 28 | $\mathrm{Sr}^+\left(4p^66g\ ^2\mathrm{G}_{7/2}\right) + \mathrm{H}(1s\ ^2\mathrm{S}_{1/2})$ | 9.514262 |
| 29 | $\mathrm{Sr}^+\left(4p^66g\ ^2\mathrm{G}_{9/2}\right) + \mathrm{H}(1s\ ^2\mathrm{S}_{1/2})$ | 9.514262 |
| 30 | $\mathrm{Sr}^+\left(4p^69s\ ^2\mathrm{S}_{1/2}\right) + \mathrm{H}(1s\ ^2\mathrm{S}_{1/2})$ | 9.653112 |
| 31 | $\mathrm{Sr}^{2+}\left(4p^6\ ^1\mathrm{S}_0\right) + \mathrm{H}^-(1s^2\ ^1\mathrm{S}_0)$ | 10.2762764 |

**Table A2.** SrH (k $^2\Sigma^+$) molecular states, the corresponding scattering channels, their asymptotic energies with respect to the ground-state level (taken from NIST [31]).

| k | Scattering Channels | Asymptotic Energies (eV) |
|---|---|---|
| 1 | $\mathrm{Sr}(5s^2\ ^1\mathrm{S}) + \mathrm{H}(1s\ ^2\mathrm{S})$ | 0.0 |
| 2 | $\mathrm{Sr}(5s5p\ ^3\mathrm{P}^\circ) + \mathrm{H}(1s\ ^2\mathrm{S})$ | 1.8228877 |
| 3 | $\mathrm{Sr}(5s4d\ ^3\mathrm{D}) + \mathrm{H}(1s\ ^2\mathrm{S})$ | 2.2631734 |
| 4 | $\mathrm{Sr}(5s4d\ ^1\mathrm{D}) + \mathrm{H}(1s\ ^2\mathrm{S})$ | 2.4982425 |
| 5 | $\mathrm{Sr}(5s5p\ ^1\mathrm{P}^\circ) + \mathrm{H}(1s\ ^2\mathrm{S})$ | 2.6902652 |
| 6 | $\mathrm{Sr}(5s6s\ ^3\mathrm{S}) + \mathrm{H}(1s\ ^2\mathrm{S})$ | 3.600349 |
| 7 | $\mathrm{Sr}(5s6s\ ^1\mathrm{S}) + \mathrm{H}(1s\ ^2\mathrm{S})$ | 3.7929029 |
| 8 | $\mathrm{Sr}(4d5p\ ^3\mathrm{F}^\circ) + \mathrm{H}(1s\ ^2\mathrm{S})$ | 4.1725763 |
| 9 | $\mathrm{Sr}(4d5p\ ^1\mathrm{D}^\circ) + \mathrm{H}(1s\ ^2\mathrm{S})$ | 4.1940009 |
| 10 | $\mathrm{Sr}(5s6p\ ^3\mathrm{P}^\circ) + \mathrm{H}(1s\ ^2\mathrm{S})$ | 4.2061469 |
| 11 | $\mathrm{Sr}(5s6p\ ^1\mathrm{P}^\circ) + \mathrm{H}(1s\ ^2\mathrm{S})$ | 4.2276633 |
| 12 | $\mathrm{Sr}(5s5d\ ^1\mathrm{D}) + \mathrm{H}(1s\ ^2\mathrm{S})$ | 4.3056546 |
| 13 | $\mathrm{Sr}(5s5d\ ^3\mathrm{D}) + \mathrm{H}(1s\ ^2\mathrm{S})$ | 4.3431318 |
| 14 | $\mathrm{Sr}(5p^2\ ^3\mathrm{P}) + \mathrm{H}(1s\ ^2\mathrm{S})$ | 4.4051164 |
| 15 | $\mathrm{Sr}(4d5p\ ^3\mathrm{D}^\circ) + \mathrm{H}(1s\ ^2\mathrm{S})$ | 4.5181299 |
| 16 | $\mathrm{Sr}(5s7s\ ^3\mathrm{S}) + \mathrm{H}(1s\ ^2\mathrm{S})$ | 4.6400683 |
| 17 | $\mathrm{Sr}^+(5s\ ^2\mathrm{S}) + \mathrm{H}^-(1s^2\ ^1\mathrm{S})$ | 4.9408674 |

## Notes

1   http://kurucz.harvard.edu/atoms/3801/ (accessed on 1 October 1996).

2   http://marcs.astro.uu.se (accessed on 1 August 2019).

3   https://physics.nist.gov/PhysRefData/ASD (accessed on 1 October 1996).

4   http://www.inasan.ru/~lima/ (accessed on 21 August 2021).

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
