# Peer review of "Inelastic Processes in Strontium-Hydrogen Collisions and Their Impact on Non-LTE Calculations"

_atoms, doi:10.3390/atoms10010033_

Round 1

Reviewer 1 Report

This is a well-done paper dealing with the effects of Sr collisions with H on non-LTE calculations relevant to interpretations in astrophysics. The authors have done a more accurate job of calculating the collision rates than were done previously.  The write-up is reasonably clear.  I have two suggestions for the authors which might improve the clarity of the manuscript:

1.  A few sentences on the calculational method might be helpful.  Calling the method a "multichannel quantum mechanical approach"  does not, in my judgment, give enough detail for the reader to assess the methodology.

2.  The authors use very old methodologies to estimate the photoionization cross sections of excited states and some of thesecould be somewhat inaccurate.  Can they give some idea of the dependence of their results on these cross sections?

Author Response

We thank the Referee for the report and for the recomendation to publish our manuscript after proper reply on the Referee comments. The attached are Referee comments and our responses.

Reviewer 2 Report

see attached file

Author Response

(The authors gave the same response as above.)

Round 2

Reviewer 2 Report

see attached file

Author Response

The second referee in the second referee report wrote the only comment: "Once the s and r processes are satisfactorily explained in the manuscript (clearly stating what exactly slow and rapid refers to) in my opinion it is acceptable for publication in Atoms."

We added a text (highlighted by the bold face) in the beginning of the Introduction (lines 14-21) about the rapid and slow processes, in particular, defining explicitly what is a rapid process and what is a slow one as recommended by the referee.

In addition, we extended a bit the end of the Introduction for clarity, as well as went through the whole MS to improve English and to present better the results and their discussion, for example, writing a number of the partial processes studied in the present paper: 1202 processes in total.